# RefMask3D: Language-Guided Transformer for 3D Referring Segmentation

## ABSTRACT

3D referring segmentation is an emerging and challenging vision-language task that aims to segment the object described by a natural language expression in a point cloud scene. The key challenge behind this task is vision-language feature fusion and alignment. In this work, we propose RefMask3D to explore the comprehensive multi-modal feature interaction and understanding. First, we propose a Geometry-Enhanced Group-Word Attention to integrate language with geometrically coherent sub-clouds through cross-modal group-word attention, which effectively addresses the challenges posed by the sparse and irregular nature of point clouds. Then, we introduce a Linguistic Primitives Construction to produce semantic primitives representing distinct semantic attributes, which greatly enhance the vision-language understanding at the decoding stage. Furthermore, we introduce an Object Cluster Module that analyzes the interrelationships among linguistic primitives to consolidate their insights and pinpoint common characteristics, helping to capture holistic information and enhance the precision of target identification. The proposed RefMask3D achieves new state-of-the-art performance on 3D referring segmentation, 3D visual grounding, and also 2D referring image segmentation. Especially, RefMask3D outperforms previous state-of-the-art method by a large margin of **5.36% mIoU** on the challenging ScanRefer dataset.

## CCS CONCEPTS

• **Computing methodologies** → **Scene understanding**.

## KEYWORDS

3D referring segmentation, Language-guided Transformer, vision-language learning

## 1 INTRODUCTION

Given a point cloud scene and a natural language description of the target object within the scene, 3D referring segmentation [16, 35, 40] aims at predicting a point-wise mask for the target object. Despite its similarity to 3D visual grounding task [2, 5] that focuses on predicting bounding boxes based on language descriptions, 3D referring segmentation has remained underexplored. This disparity in research attention could be attributed to the fact that 3D referring segmentation demands point-wise masks and a deeper, nuanced understanding of intricate fine-grained semantics.

*ACM MM, 2024, Melbourne, Australia*

© 2024 Copyright held by the owner/author(s). Publication rights licensed to ACM.
ACM ISBN 978-x-xxxx-xxxx-x/YY/MM
https://doi.org/10.1145/nnnnnnn.nnnnnnn

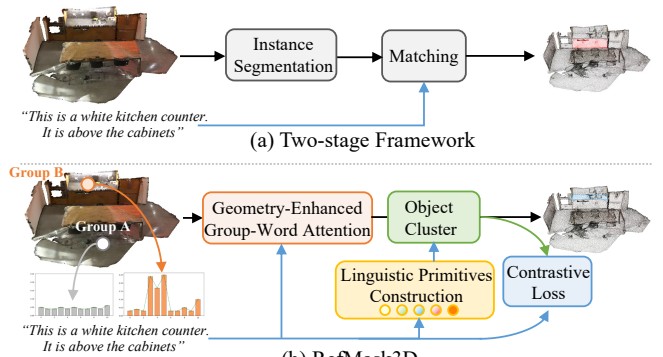

**Figure 1: (a) Two-stage framework, fusing language features in the later matching stage, exhibit limited interactions and weak alignment between vision and language features. In contrast, (b) our RefMask3D conducts comprehensive vision-language fusion in both the early feature encoding stage and decoding stage. Combined with contrastive learning, our model learns a well-structured vision-language joint feature space than two-stage methods.**

The prevailing methods [2, 5, 16, 35] in 3D referring segmentation, as shown in Figure 1 (a), typically adopt a two-stage *segmentation-then-matching* pipeline that firstly segments redundant objects using a pre-trained instance segmentation model, and then selects the target object by matching visual and linguistic features [16, 35]. However, this pipeline exhibits inherent limitations. Given that the matching phase relies on segmentation predictions from the first phase, any omissions or inaccuracies at the first phase inevitably significantly weaken the accuracy of the following matching phase. Furthermore, simplistic incorporation of textual information during matching fails to analyze the semantic impact of individual words, thus overlooking complex semantic nuances and leading to matching errors. These issues underscore the necessity for a unified approach that integrates both segmentation and matching processes with a deeper linguistic understanding. In response, our work introduces a streamlined, effective end-to-end pipeline that comprehensively leverages language information. The proposed method, **RefMask3D**, enhances the interaction and understanding between vision and language, aiming to enhance the performance of 3D referring segmentation.

First, different from previous two-stage works [2, 5, 16, 35] that adopt *segmentation-then-matching* pipeline, we propose to fully harness the early feature encoding layers to extract rich multi-modal context. To this end, we introduce a Geometry-Enhanced Group-Word Attention, which conducts cross-modal attention between language and local groups (sub-clouds) with geometrically adjacent points at each stage of the Point Encoder. This integration

not only reduces the noise from direct point-to-word correlations, which is common due to the sparse and irregular nature of point clouds, but also leverages the intrinsic geometric relationships and fine-grained 3D structure within the point clouds. The proposed Geometry-Enhanced Group-Word Attention significantly improves the model's ability to interact with and understand linguistic and geometric data. Additionally, we incorporate a learnable "background" language token to prevent the entanglement of irrelevant language features with local group features. For example, as shown in Figure 1 (b), "Group A" as a non-target group exhibits low responses to these irrelevant words. This approach ensures that point features are enriched with semantic-related linguistic information, maintaining a continuous and context-aware awareness of the relevant language context at each group or point in the network.

Second, with the fused vision-language features, we further design how to effectively identify the target object in decoder. Effectively localizing the target significantly hinges on the cues provided by the query input to the cross-modal decoder [11, 29, 31]. A majority of existing works [18, 33, 37, 41] initialize non-parametric query with point positions sampled with furthest point sampling (FPS) or parametric learnable query. Such strategies typically aim to scan the whole point cloud scene to identify all possible objects as candidates. However, it inadvertently sidesteps an essential aspect: the fact that only one language-referred target object is present, potentially leading to insufficient training or posing challenges in optimization. Although 3D-SPS [31] proposes to select the top-$k$ query points close to the given language description to alleviate this issue, the selection process is susceptible to noise, particularly in scenarios where the chosen points do not encompass the language-related target. To address these challenges, we introduce a strategy termed Linguistic Primitives Construction (LPC). We initialize a diverse set of primitives, each designed to represent distinct semantic attributes such as shape, color, size, relationships, location, and so on. By engaging in interactions with specific linguistic information, these primitives are capable of acquiring corresponding attributes. Feeding these semantically enriched primitives into the decoder enhances the network's focus on diverse semantics within the point cloud, thereby significantly improving the model's ability to accurately localize and identify the target object.

Furthermore, an Object Cluster Module is proposed to capture holistic information and generate the object embedding for segmenting the target object. Linguistic primitives are designed to focus on specific parts of a point cloud that correlate with their semantic attributes. However, our ultimate goal is to identify a unique target object based on the given text, which requires a holistic understanding of language. To achieve this, we propose an Object Cluster Module. This module first explores the relationships among the linguistic primitives to discern commonalities and differences in their focus areas. Utilizing this insight, we initialize language-based queries to capture these common characteristics, which form the final object embedding crucial for identifying the target object. The proposed Object Cluster Module greatly help the model to deepen the holistic understanding of linguistic and visual information.

Our main contributions are summarised as follows:

- We propose a Geometry-Enhanced Group-Word Attention, which enhances cross-modal interactions by integrating language with geometrically coherent sub-clouds, effectively addressing the challenges posed by the sparse and irregular nature of point clouds.
- We design a Linguistic Primitives Construction, a strategy that learns primitives to represent distinct semantic attributes, enhancing the model's capability to accurately identify targets through interaction with specific linguistic information.
- We introduce an Object Cluster Module that analyzes the interrelationships among linguistic primitives to unify their insights and pinpoint common characteristics to deepen the holistic understanding of linguistic and visual information.
- We achieve new state-of-the-art performance on 3D referring segmentation and visual grounding, and significantly outperform previous methods by a large margin, *e.g.*, **5.36% mIoU** on the challenging ScanRefer dataset.

## 2 RELATED WORKS

### 2.1 3D Instance and Referring Segmentation

3D Instance Segmentation aims to detect and segment instances in the sparse point clouds. The success of the transformer has permeated the 3D modeling domain. In 3D object detection, methods [29, 33] employing the transformer have achieved state-of-the-art results. Inspired by this, transformer-based methods like OneFormer3D [23], Mask3D [37], SPFormer [38], MAFT [24] for instance segmentation have subsequently been developed. Mask3D [37] treats each object as an instance query. Through Transformer decoders, it learns these queries by attending to multi-scale point cloud features and the queries concurrently produce all instance masks integrating with point features.

Influenced by advancements in 2D multimodality, 3D referring segmentation is increasingly gaining attention. This field focuses on segmenting a target instance based on a given language expression. TGNN [16] is the first work to solve this challenging problem and introduces aggregating textual features by considering the neighboring local structure of each instance but also emphasizes capturing spatial interactions centered around each object. X-RefSeg3D [35] follows the paradigm of TGNN and builds a multi-modal graph for the given 3D environment, integrating textual and spatial connections to facilitate reasoning through the use of graph neural networks. 3D-STMN [40] is proposed to construct dense superpoint-text matching which is enhanced through dependency-driven insights in the end-to-end paradigm. Despite the notable achievements of existing methods, they fall short in facilitating comprehensive multi-modal feature interaction and understanding. This paper is dedicated to addressing this challenge.

### 2.2 3D Visual Grouding

3D visual grounding aims to identify and locate the referred object in a 3D scene based on the language expression. Three datasets including ScanRefer [5], ReferIt3D [2] and Multi3DRefer [47] are proposed to facilitate the research which contains object-expression pairs based on ScanNet [9]. Previous methods [1, 3, 12–15, 17, 36, 44, 44, 46, 48] mostly employ a two-stage pipeline. They first exploit either a 3D object detector [20, 22, 29] or ground truth information to generate object proposals. Subsequently, they utilize a text encoder [10, 27] to extract linguistic features and then identify

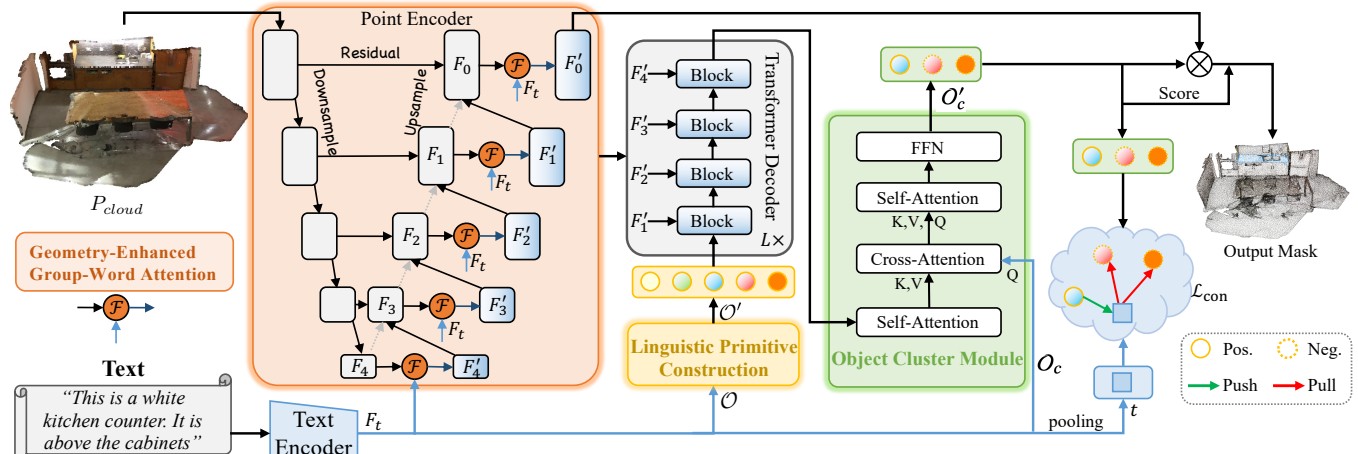

**Figure 2: The framework overview of the proposed RefMask3D. It extracts text-enriched point features from the point encoder which is assisted by Geometry-Enhanced Group-Word Attention. Subsequently, the Linguistic Primitives Construction Module generates primitives to embody specific semantic attributes. These primitives are then fed into the Transformer Decoder to focus on diverse semantics. Object Cluster Module is employed to analyze the interrelationships among linguistic primitives to unify their insights and pinpoint common characteristics to enhance the precision of target identification.**

the referred object through feature fusion and language feature matching. In this context, InstanceRefer [46] streamlines the task by approaching it as an instance-matching challenge. LanguageRefer [36] turns the multi-modal problem into a language modeling scenario, achieved by substituting 3D features with predicted object labels. SAT [44] employs additional 2D semantics to boost multi-modal alignment and achieve superior performance.

In contrast to the two-stage methods, 3D-SPS [31] adopts a progressive keypoint selection strategy guided by language and locates the target directly within a single-stage network for the first time. BUTD-DETR [18] calculates the correlation between each word and object, subsequently selecting the word features that align with the object's name to match the corresponding object. EDA [41] builds upon BUTD-DETR and further proposes text decoupling to separate text components by grammatical analysis. Single-stage methods have taken a dominant role and both BUTD-DETR and EDA have demonstrated superior performance. However, these methods either need to train an additional text span predictor network or employ external tools to decouple text which is time-consuming and complicated. Besides, they are designed for visual grounding and cannot perfectly adapt to referring segmentation. To this end, we propose our method to simplify the image-text matching and apply it to both 3D referring segmentation and visual grounding.

## 3 METHODS

### 3.1 Architecture Overview

Figure 2 shows the overall architecture of our proposed end-to-end 3D referring segmentation approach **RefMask3D**. RefMask3D takes an input pair comprising a point cloud scene and a textual description, and produces a point-wise mask for the target object indicated by the textual description. The point cloud scene, denoted as $P_{cloud} \in \mathbb{R}^{N \times (3+F)}$, consists of a total of $N$ points, with each

point containing 3D coordinate information $\in \mathbb{R}^3$ and an auxiliary feature $\in \mathbb{R}^F$ like color.

First, a text encoder is employed to embed the text description into language features, denoted as $F_t \in \mathbb{R}^{N_t \times D}$, where $D$ and $N_t$ represent the number of channels and words, respectively. We extract point features from the point encoder which builds deep interaction between vision and language through Geometry-Enhanced Group-Word Attention. The point encoder is a 3D U-Net-like backbone. In the vanilla U-Net architecture, the feature map at the $i$-th layer in the upsampling path is obtained by combining the features from the corresponding $i$-th layer in the downsampling path with the features from the $(i + 1)$-th layer which is illustrated in the gray line in Figure 2. The vanilla output features denote as $F_i \in \mathbb{R}^{N_i \times D}, i = \{1, 2, 3, 4\}$. Therefore, the multi-scale vision-language features derived from the point encoder are denoted as $F'_i \in \mathbb{R}^{N_i \times D}, i = \{1, 2, 3, 4\}$. The full-resolution feature map $F'_0 \in \mathbb{R}^{N \times D}$ is used as the mask feature for mask predictions.

Then, Linguistic Primitives Construction produces primitives $O'$ to represent distinct semantic attributes by informative language cues, enhancing the model's capability to accurately localize and identify targets through interaction with specific linguistic information. The linguistic primitives $O'$, the multi-scale point features $\{F'_1, F'_2, F'_3, F'_4\}$, and language features $F_t$ collectively serve as input to a 4-layer cross-modal transformer decoder, which is applied $L$ times, leading to an overall transformer decoder with $4L$ layers.

Next, the output of linguistic primitives through transformer decoder and object queries $O_c$ are fed into the Object Cluster Module to analyze the interrelationships among linguistic primitives to unify their insights and pinpoint common characteristics. Finally, we obtain masks by performing a multiplication operation between the last-layer object embedding $O'_c$ and mask feature $F'_0$. Meanwhile, we employ a MLP head to predict confidence score. The mask with the highest confidence score is selected as the output.

## 3.2 Geometry-Enhanced Group-Word Attention

Different from models [16, 18, 31, 41] that stack a modality fusion module on top of the vision or language backbones, our approach integrates multi-modal fusion within the point encoder, leveraging the advantages of an end-to-end paradigm. Early-stage fusion of cross-modal features is believed to enhance the effectiveness of the fusion process in 2D-RES tasks [43]. However, it remains unexplored within the 3D field. The 3D domain, characterized by the sparse and irregular distribution of point cloud data, presents unique challenges to cross-modal fusion. The conventional 2D method of directly establishing relationships between each point and linguistic terms introduces excessive noise, potentially impairing performance due to the complexity and variability of point cloud structures. Addressing this challenge, we introduce Geometry-Enhanced Group-Word Attention mechanism. Unlike traditional methods that calculate cross-modal relationships at the individual point level, our approach innovatively processes local groups (sub-clouds) with geometrically adjacent points. This methodology not only mitigates the noise associated with direct point-to-word correlations but also capitalizes on the inherent geometric relationships within point clouds, enhancing the model's ability to accurately integrate linguistic and 3D structure.

Firstly, we employ farthest point sampling (FPS) to downsample the point number of features $F_i$ from $N_i$ to $N_g$, which serves as a set of local centroids. Following the identification of these local centroids, we proceed to gather their neighboring points. This is achieved through the application of the $k$-nearest Neighbor ($k$-NN) algorithm, which allows us to systematically associate each local centroid with its adjacent points based on its geometric proximity. The aforementioned process is formulated as:

$$F_i^g = k\text{-NN}(\text{FPS}(F_i), F_i) \in \mathbb{R}^{N_g \times k \times D}. \tag{1}$$

As such, we obtain $N_g$ local groups, each of which consists of $k$ points. Next, we perform group-word cross-modal attention between $N_g$ local groups and language features $F_t$, We formulate it as:

$$S_i^g = \frac{F_i^g F_t^T}{\sqrt{D}} \in \mathbb{R}^{N_g \times k \times N_t}, \tag{2}$$

where $S_i^g$ denotes the relationship between local groups $F_i^g$ and $F_t$ and we sum along $k$ nearest neighbors dimention to obtain the relationship between each local centroid with $F_t$ denoted as $S_i^c \in \mathbb{R}^{N_g \times N_t}$. Subsequently, we extract linguistic features $F_{it}^c$ related to the local centroids as,

$$F_{it}^c = \text{softmax}(S_i^c)F_t \in \mathbb{R}^{N_g \times D}. \tag{3}$$

Finally, inspired by PointNet++ [34], we propagate $F_{it}^c$ from each local centroid to its corresponding original point number with a weighted summation and obtain linguistic features related to the whole point feature map $F_{it} \in \mathbb{R}^{N_i \times D}$ which have the same shape as $F_i$. We combine them to generate multi-modal feature maps $F_i'$ via element-wise multiplication.

$$F_i' = F_{it} \odot F_i. \tag{4}$$

Through the above process, we have implemented cross-modal fusion based on the sparse and irregular characteristics of 3D data in the point feature extraction stage.

Besides, in vanilla cross-modal attention, the challenge arises when dealing with situations where a point has no corresponding words related to it. To tackle this, we introduce learnable background embeddings for language features, denoted as $T_{bg} \in \mathbb{R}^D$. This strategy is designed to allow points without corresponding text information to focus on a generic background text embedding, $T_{bg}$, reducing the potential distortion caused by unrelated text on the point feature. Specifically, at each feature encoding layer $i$, we perform a concatenation operation on the language feature $F_t$ with the background feature $T_{bg}$. Subsequently, the above relationship calculation process becomes:

$$S_i^{g'} = \frac{F_i^g [F_t; T_{bg}]^T}{\sqrt{D}},$$
$$F_{it}^{c'} = \circledR(\text{softmax}(S_i^{c'}))F_t, \tag{5}$$

where $[;]$ represents the concatenation operation, and $D$ is the channel dimension. $S_i^{g'}$ enables point features without language element correspondence to emphasize the background feature, reducing irrelevant feature impact. The operation $\circledR(\cdot)$ is used to remove the last column from a given matrix, which prevents the integration of the background feature into the final fused feature. As such, we obtain $F_{it}^{c'}$ which represent refined linguistic features related to the local centroids and not influenced by irrelevant words. The background embedding is a learnable parameter, designed to capture the overall data distribution of the dataset and represent background information effectively. This embedding is only used during the attention calculation process. By incorporating the background embedding, we facilitate more accurate cross-modal interactions, which are not adversely influenced by irrelevant words.

## 3.3 Linguistic Primitives Construction

In the 3D domain, the existing approaches typically initialize queries with point coordinates sampled from the input point cloud [29, 33, 37]. These sampling strategies are based on either the farthest point sampling method [33, 37] or the $k$-closest points sampling technique [18, 29]. Such methods are prevalent in tasks like visual grounding [18, 29, 41] and instance segmentation [37]. However, a key limitation of these methods is their neglect of language information, which is vital for precise referring segmentation. Relying solely on farthest point sampling often results in predictions that stray from the objects of interest, especially in sparse scenes, thereby hindering convergence or resulting in the loss of objects. Although 3D-SPS [31] attempts to address this issue by introducing a language-aware query selection that selects the top-$k$ query points close to the given language description. Yet, this approach is manually tailored and susceptible to noise. This becomes particularly problematic when the selected points don't accurately reflect the target object or when they are all anchored to a single word. In response to these challenges, we propose a Linguistic Primitives Construction to incorporate semantic content from language that learn diverse linguistic primitives to locate objects related to corresponding semantic properties separately.

In the process outlined in Figure 3, we begin by initializing $N_o$ learnable primitives by sampling them from different Gaussian distributions, each specifying a distinct semantic property to be

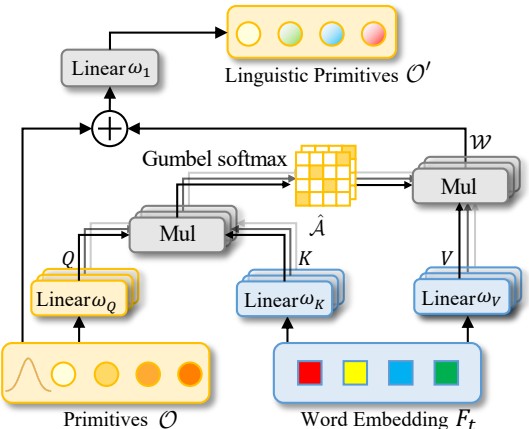

**Figure 3: Linguistic Primitives Construction (LPC) initializes various primitives $O$ to express distinct semantic attributes. These primitives after interacting with linguistic information are capable of acquiring corresponding attribute values denoted $O'$.**

captured. represented by $O$, to serve as our primitives,

$$O_i \sim \mathcal{N}(\mu_i, (\sigma_i)) \in \mathbb{R}^{N_o \times D}, \tag{6}$$

where $N_o$ is the number of primitives, $\mu_i$ and $\sigma_i$ are learnable parameters of the Gaussian distribution. These primitives are assumed to contain diverse semantics *e.g.*, shape, color, size, material, relationship, and location. Then, we let each primitive aggregate language of distinct linguistic entities and extract corresponding information from the given language. For example, in the description in Figure 2, the primitive responsible for color concentrates for "white", and the primitive responsible for location highlight "above". To achieve this, three distinct linear layers, $\omega_Q$, $\omega_K$, and $\omega_V$, are employed to project the primitives and the language word feature $F_t$ into a shared feature space, which are represented as $Q$, $K$, and $V$, respectively. Subsequently, we calculate the similarity between each pair of primitive and word feature, formulated as:

$$\mathcal{A} = \frac{QK^T}{\sqrt{D}} \in \mathbb{R}^{N_o \times N_t}. \tag{7}$$

After computing the similarity matrix $\mathcal{A}$, unlike the conventional cross-attention that performs a softmax operation to normalize the correlation of all words corresponding to a specific primitive, we aim to enable each primitive to concentrate on the most related semantic. *e.g.* "white". Therefore, we employ the differentiable Gumbel-Softmax approach [19, 42] to assign the word features to the primitives $Q$ in a differentiable way and generate the corresponding clustering matrix $\hat{\mathcal{A}}$ between $Q$ and $K$,

$$\hat{\mathcal{A}} = \text{Gumbel-Softmax}(\mathcal{A}) \in \mathbb{R}^{N_o \times N_t}. \tag{8}$$

With $\hat{\mathcal{A}}$, each primitive effectively chooses the word embeddings corresponding to its semantic and then obtains linguistic primitives $O'$, which are formulated by:

$$O' = \omega_1(\hat{\mathcal{A}}V), \tag{9}$$

where $\omega_1$ is the linear projection layer. The linguistic primitives are designed to exhibit semantic patterns. Feeding such primitives into the transformer decoder enables it to emphasize diverse language information, which contributes to accurately identifying target objects in the later stage.

### 3.4 Object Cluster Module

Each linguistic primitive focuses on different semantic patterns in the given point cloud that correlate with their linguistic attributes. However, our ultimate goal, based on the given text, is to identify a unique target object, which requires a comprehensive understanding of language. For example, the primitive responsible for "white" will focus on all the white parts of the picture, but we need to find the one above the cabinets. To this end, we employ an Object Cluster Module to explore the relationships among the linguistic primitives to identify commonalities and differences in their focus areas. Specifically, we initialize $N_c$ potential object queries $O_c \in \mathbb{R}^{N_c \times D}$, leveraging sentence-level linguistic features. This initialization facilitates a deeper grasp of object descriptions. We conduct a self-attention mechanism to extract these common characteristics from linguistic primitives. In the decoding process, we input these object queries, treating them as queries, with the common characteristics enriched linguistic primitives acting as keys/values. This setup enables the decoder to merge linguistic insights from linguistic primitives into object queries, efficiently identifying and grouping the referred object's queries into $O_c'$, thereby achieving precise object identification.

### 3.5 Training Objectives

While the proposed Object Cluster Module aids in identifying target objects, it does not effectively mitigate ambiguities arising from other embeddings. These ambiguities could lead to false positives during the inference stage. We employ contrastive learning to differentiate the target token from others. This is achieved by maximizing the similarity between the target token and the corresponding expression while minimizing similarities with non-target (negative) pairs. The process can be formulated as,

$$\mathcal{L}_{con} = -\log \frac{\exp(<t, o_+ > / \tau)}{\sum_{k=1}^{N_o} \exp(<t, o_k > / \tau)}, \tag{10}$$

where $<,>$ denotes cosine similarity, $\tau$ is the temperature parameter, $t \in \mathbb{R}^D$ represents the text embedding obtained by pooling language features $F_t$. The set $\{o_k\}_{k=1}^{N_o}$ comprises object embeddings derived from the output of the object cluster decoder $O_c'$, with $o_+$ being positive object embedding that matches with ground-truth mask.

During the training phase, we use Hungarian matching to select an object embedding with the lowest cost to the ground-truth object as our output. The matching losses include single-class cross-entropy loss on the output score, binary cross-entropy loss, and dice loss on mask prediction,

$$\mathcal{L}_{match} = \lambda_{cls}\mathcal{L}_{cls} + \lambda_{bce}\mathcal{L}_{bce} + \lambda_{dice}\mathcal{L}_{dice}. \tag{11}$$

The total training loss is calculated as: $\mathcal{L} = \mathcal{L}_{match} + \lambda_{con}\mathcal{L}_{con}$, where the weights $\lambda_{cls}$, $\lambda_{bce}$, $\lambda_{dice}$, and $\lambda_{con}$ are used for balancing different loss terms.

**Table 1: Ablation study of the proposed method on ScanRefer dataset. GEGWA, LPC, and OCM denote Geometry-Enhanced Group-Word Attention, Linguistic Primitives Construction, and Object Cluster Module, respectively.**

| Components | | | Results | |
|---|---|---|---|---|
| GEGWA | LPC | OCM | mIoU | Acc@0.5 |
| ✗ | ✗ | ✗ | 39.24 | 43.95 |
| ✓ | ✗ | ✗ | 42.16 (+2.92) | 46.56 (+2.61) |
| ✗ | ✓ | ✗ | 41.58 (+2.34) | 45.81 (+1.86) |
| ✗ | ✗ | ✓ | 40.81 (+1.57) | 44.97 (+1.02) |
| ✓ | ✓ | ✗ | 43.83 (+4.59) | 48.15 (+4.20) |
| ✓ | ✗ | ✓ | 43.12 (+3.88) | 47.48 (+3.53) |
| ✗ | ✓ | ✓ | 42.75 (+3.51) | 47.02 (+3.07) |
| ✓ | ✓ | ✓ | **44.86** (+5.62) | **49.24** (+5.29) |

**Table 2: Ablation studies of different fusion strategies for GEGWA on ScanRefer.**

| Encoder | mIoU | Acc@0.5 |
|---|---|---|
| No fusion | 39.24 | 43.95 |
| Point-level fusion | 40.33 | 44.78 |
| Ours w/o bg | 41.25 | 45.62 |
| **Ours** | **42.16** | **46.56** |

## 4 EXPERIMENTS

### 4.1 Datasets and Evaluation Metrics

The *ScanRefer* dataset [5] is based on the 800 *ScanNet* scenes [9] and comprises 51,583 linguistic descriptions. On average, each scene contains 13.81 objects and 64.48 descriptions. The dataset's performance is evaluated using the Acc@$m$IoU metric. It represents the proportion of descriptions where the predicted box or mask aligns with the ground truth, having an IoU > $m$, where $m \in \{0.25, 0.5\}$. The results are categorized into *unique* and *multiple*. An object is regarded as *unique* if it's the sole entity of its class in a scene, and *multiple* otherwise. For 3D referring segmentation, there is another metric mean intersection-over-union (mIoU) which calculates the IoU between the prediction and ground truth masks averaged across all the test samples.

### 4.2 Implementation Details

Our experimental setup generally adheres to the default configurations of Mask3D [37], except where explicitly noted. For the point encoder, we utilize Minkowski Res16UNet34C [8]. For language embeddings, we utilize BERT [10], following its proven effectiveness in capturing linguistic nuances. We employ the AdamW optimizer [21] with an initial learning rate of $4 \times 10^{-5}$, accompanied by a cosine decay schedule to adjust the learning rate progressively. We configure the training process to span a maximum of 20 epochs with a batch size of 16. Following Mask3D, we set the coefficients for different loss functions $\lambda_{cls}$, $\lambda_{bce}$, and $\lambda_{dice}$ to 2, 5, and 5, respectively. For hyper-parameter optimization, we meticulously choose values that best suit our model's architecture and the task's complexity. Specifically, we set the contrastive loss coefficient $\lambda_{con}$ to 0.3, the temperature parameter $\tau$ to 0.05 to manage the softmax distribution's sharpness, and the layer count $L$ to 3. Additionally,

**Table 3: Ablation studies of different groups and neighbors numbers for GEGWA on ScanRefer.**

| $N_g$ | $k$ | mIoU | Acc@0.5 |
|---|---|---|---|
| - | - | 39.24 | 43.95 |
| 64 | 8 | 41.54 | 45.86 |
| 64 | 32 | 42.25 | 46.63 |
| 32 | 16 | 41.18 | 45.47 |
| 128 | 16 | 42.39 | 46.78 |
| **64** | **16** | 42.16 | 46.56 |

**Table 4: Ablation studies of input query designs for Linguistic Primitives Construction (LPC) on ScanRefer.**

| Input Query | mIoU | Acc@0.5 |
|---|---|---|
| FPS | 39.24 | 43.95 |
| Top-$k$ | 40.18 | 44.31 |
| Ours w/o Gaussian | 40.51 | 44.74 |
| Ours w/o Gumbel-softmax | 40.86 | 45.03 |
| **Ours** | **41.58** | **45.81** |

we determine the number of primitives $N_o$ to 100 and the potential object queries $N_c$ to 10, ensuring the model's capacity to handle diverse and intricate object interactions effectively.

### 4.3 Ablation Study

• **Component Analysis.** We conduct detailed experiments to demonstrate the effectiveness of each component in the proposed method. As shown in Table 1, our vanilla baseline integrating Mask3D with language input only in the transformer decoder achieves a mIoU of 39.24%, establishing a strong pipeline. Then, the introduction of the Geometry-Enhanced Group-Word Attention (**GEGWA**) enhances performance by 2.92% mIoU, highlighting its effectiveness in fusing linguistic and visual features in the point encoder and minimizing the influence of irrelevant points and words for improved cross-modal interaction. Furthermore, incorporating the Linguistic Primitives Construction (**LPC**) boosts mIoU by 2.34%, underscoring its importance in clustering diverse semantic language information for target identification in 3D referring segmentation. The addition of the Object Cluster Module (**OCM**) further enhances our model, contributing a 1.57% increase in mIoU by constructing a discriminative identification for the target. By combining all these components in **RefMask3D**, we achieve new state-of-the-art performance with a remarkable mIoU of 44.86% and an Acc@0.5 of 49.24% on the challenging ScanRefer dataset, demonstrating the effectiveness of our proposed approach.

• **Different Fusion Strategies for GEGWA.** In Table 2, we evaluate the impact of different fusion designs. The absence of vision-language fusion within the encoder results in a decrease of 2.92% in mIoU, confirming the effectiveness of early-stage fusion. The w/o bg configuration uses a standard cross-attention module without background embeddings to filter out irrelevant information. This approach underperforms our method by 0.91% in mIoU, highlighting the importance of eliminating irrelevant data in cross-modal interactions. The point-level fusion setup directly conducts fusion in the individual point without considering the neighbors and geometric information, leading to a 1.83% reduction in mIoU. This

**Table 5: 3D referring expression segmentation benchmark results on ScanRefer, evaluated by mIoU, accuracy IoU 0.25 and IoU 0.5. † The accuracy is obtained through our own implementation, which involved adding an auxiliary mask head.**

| Stage | Method | Reference | Modality | Unique (~19%) 0.25 | 0.5 | Multiple (~81%) 0.25 | 0.5 | Overall 0.25 | 0.5 | mIoU |
|---|---|---|---|---|---|---|---|---|---|---|
| Two | TGNN [16] | AAAI'21 | 3D | - | - | - | - | 37.50 | 31.40 | 27.80 |
| | X-RefSeg3D [35] | AAAI'24 | 3D | - | - | - | - | 40.33 | 33.77 | 29.94 |
| Single | BUTD-DETR [18]† | ECCV'22 | 3D | 76.63 | 63.30 | 38.01 | 29.70 | 45.53 | 36.22 | 35.47 |
| | EDA [41]† | CVPR'23 | 3D | 79.88 | 66.42 | 40.51 | 31.24 | 48.13 | 38.06 | 36.21 |
| | 3D-STMN [40] | AAAI'24 | 3D | 89.30 | 84.00 | 46.20 | 29.20 | 54.60 | 39.80 | 39.50 |
| | **RefMask3D** (ours) | - | 3D | **89.55** | **84.69** | **48.09** | **40.77** | **55.87** | **49.24** | **44.86** |

**Table 6: 3D visual grounding results using 3D modality on ScanRefer, with IoU 0.5 as the evaluation metric.**

| Method | Reference | Unique | Multiple | Overall |
|---|---|---|---|---|
| Two-stage methods | | | | |
| ScanRefer [5] | ECCV'20 | 46.19 | 21.26 | 26.10 |
| ReferIt3D [2] | ECCV'20 | 37.50 | 12.80 | 16.90 |
| TGNN [16] | AAAI'21 | 56.80 | 23.18 | 29.70 |
| InstanceRefer [46] | ICCV'21 | 66.83 | 24.77 | 32.93 |
| SAT [44] | ICCV'21 | 50.83 | 25.16 | 30.14 |
| FFL-3DOG [12] | ICCV'21 | 67.94 | 25.70 | 34.01 |
| 3DVG-Trans. [48] | ICCV'21 | 58.47 | 28.70 | 34.47 |
| 3DJCG [4] | CVPR'22 | 61.30 | 30.08 | 36.14 |
| BUTD-DETR [18] | ECCV'22 | 64.98 | 33.97 | 38.60 |
| D3Net [6] | ECCV'22 | 70.35 | 30.50 | 37.87 |
| EDA [41] | CVPR'23 | 68.57 | 37.64 | 42.26 |
| ViewRefer [13] | ICCV'23 | - | 26.50 | 33.66 |
| Single-stage methods | | | | |
| 3D-SPS [31] | CVPR'22 | 64.77 | 29.61 | 36.43 |
| BUTD-DETR [18] | ECCV'22 | 61.24 | 32.81 | 37.05 |
| EDA [41] | CVPR'23 | 69.42 | 36.82 | 41.70 |
| **RefMask3D** (ours) | - | **78.69** | **38.15** | **45.62** |

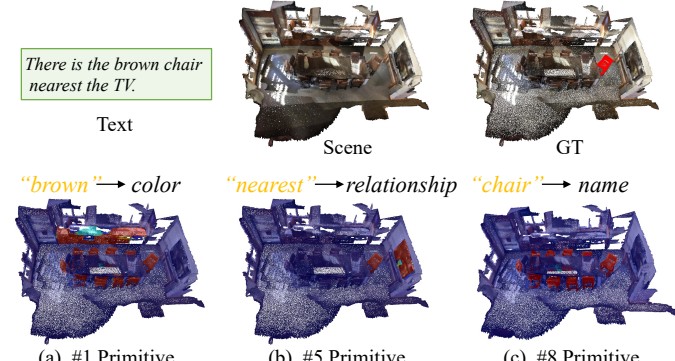

*There is the brown chair nearest the TV.*

Text      Scene      GT

*"brown"* → color    *"nearest"* → relationship    *"chair"* → name

(a) #1 Primitive      (b) #5 Primitive      (c) #8 Primitive

**Figure 4: Primitives heatmap visualization. Different primitives represent distinct semantic attributes. Blue indicates the lowest response levels, while red signifies the highest response levels.**

outcome demonstrates that processes local groups with geometrically adjacent points are beneficial for overall performance.

• **Different Group and Neighbors Numbers for GEGWA.** In Table 3, we assess the impact of varying the number of groups $N_g$ and the number of neighbors $k$. By holding $N_g$ constant at 64 and incrementally increasing $k$ from 8 to 32, we observed a rapid improvement in performance followed by a plateau. Similarly, fixing $k$ at 16 and elevating $N_g$ from 32 to 128 also resulted in a swift rise in outcomes, which then stabilized. Considering computational efficiency, we ultimately opted for $N_g = 64$ and $k = 16$ as the configuration that yields optimal results.

• **Comparison with Other Designs for LPC.** In Table 4, we examine the effect of input queries fed into the Transformer Decoder. The FPS method, widely used in point sampling and described in PointNet++ [34], covers the entire scene but lacks focus on language-relevant points. The Top-$k$ approach [31] selects the top $k$ points most relevant to a given text based on object confidence. We use $k = 100$ in our experiments. According to the results in Table 4, the proposed Linguistic Primitives Construction outperforms other query inputs, achieving promising results. Besides, we observed a notable point when not utilizing Gaussian initialization for the primitives, opting instead for learnable parameters directly, which resulted in a performance decrease of 1.07% mIoU. This finding

underscores the significance of employing Gaussian initialization to ensure independence among multiple primitives, playing a crucial role in effectively capturing diverse semantic information for the final result. Additionally, the employment of Gumbel-softmax enhances the overall performance which is in line with our analysis.

## 4.4 Comparison with State-of-the-Art Methods

• **3D Referring Segmentation Benchmark Results.** We evaluate RefMask3D on ScanRefer [5] and report the mIoU and Acc@$m$IoU performance in Table 5. For a fair comparison with SOTA methods, we implement BUTD-DETR [18] and EDA [41] with an auxiliary mask head [41]. The results show that RefMask3D outperforms existing methods, achieving the highest scores across all metrics with a significant improvement of **+5.36%** in mIoU. This demonstrates the effectiveness of RefMask3D and highlights its exceptional vision-language understanding capabilities.

• **3D Visual Grounding Benchmark Results.** Instance segmentation predictions can be readily converted into bounding box predictions by determining the minimum and maximum coordinates of the masked instances. With this capability, we extend our experimentation to the realm of 3D visual grounding. Table 6 shows the results on ScanRefer dataset [5], with Acc@0.5 as the metric. Our findings show that RefMask3D surpasses previous methods, achieving an improvement of **+3.36%** in Acc@0.5 compared to previous state-of-the-art methods. It is worth noting that this achievement

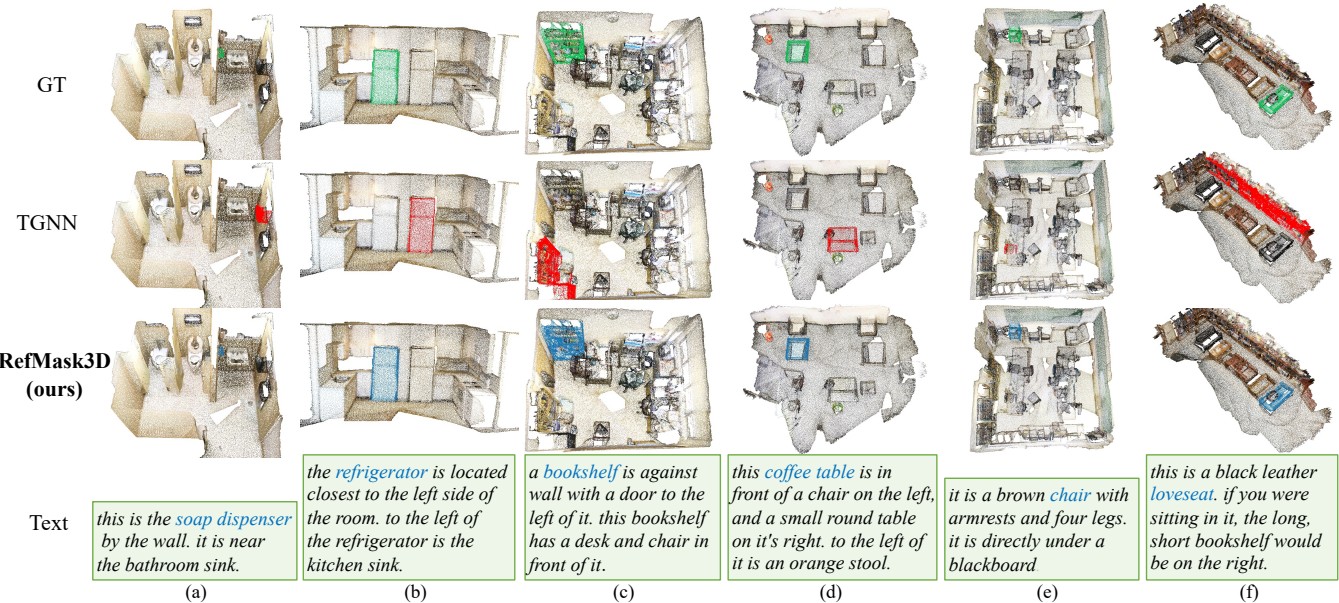

this is the *soap dispenser* by the wall. it is near the bathroom sink. (a)

the *refrigerator* is located closest to the left side of the room. to the left of the refrigerator is the kitchen sink. (b)

a *bookshelf* is against wall with a door to the left of it. this bookshelf has a desk and chair in front of it. (c)

this *coffee table* is in front of a chair on the left, and a small round table on it's right. to the left of it is an orange stool. (d)

it is a brown *chair* with armrests and four legs. it is directly under a blackboard (e)

this is a black leather *loveseat*. if you were sitting in it, the long, short bookshelf would be on the right. (f)

**Figure 5: Visualization results of complex language descriptions on ScanRefer. we use color masks for clarity: green represents the ground truth, red indicates incorrect predictions by TGNN, and blue signifies correct predictions by ours.**

**Table 7: 2D referring image segmentation benchmark results on the val split of RefCOCO/+/g. Overall IoU is employed as the evaluation metric.**

| Method | RefCOCO | RefCOCO+ | RefCOCOg |
|---|---|---|---|
| MCN [30] | 62.4 | 50.6 | 49.2 |
| CRIS [39] | 70.5 | 62.3 | 59.9 |
| RefTR [25] | 70.6 | - | - |
| LAVT [43] | 72.7 | 62.1 | 61.2 |
| VLT [11] | 73.0 | 63.5 | 63.5 |
| GRES [26] | 73.8 | 66.0 | 65.0 |
| **RefMask3D** (ours) | **75.3** | **66.9** | **66.3** |

is made using only the bounding boxes derived from instance segmentation predictions, without dedicated designs for 3D visual grounding. This highlights the effectiveness of our RefMask3D in extracting and understanding more discriminative and effective vision-language joint representations.

• **2D Referring Image Segmentation Results.** In Table 7, we apply RefMask3D to 2D referring image segmentation by using Mask2Former [7] framework based on Swin-B [28] backbone. We remove GEGWA component which is specifically designed for 3D realm. Our results are compared with the leading methods on Re-fCOCO/+/g [32, 45]. We train a single RefMask3D model on Ref-COCO/+/g without the need for extensive pre-training. RefMask3D outperforms previous methods on all three benchmarks, underscoring its versatility and effectiveness in referring segmentation.

### 4.5 Visualization

In Figure 4, we visualize the primitive heatmap on the given point cloud. Different primitives represent distinct semantic attributes

like "*color*" (a), "*relationship*" (b), "*name*" (c). The above qualitative results highlight our Linguistic Primitive Construction is capable of acquiring corresponding attribute values which improves the ability to accurately localize and identify the target object.

In Figure 5, we visualize the grounding truth masks, predictions of TGNN [16], and predictions of RefMask3D in each column from top to bottom. RefMask3D effectively interprets complex language phrases like "*soap dispenser*" (a) and accurately segments the target object, even among items of the same category (b)-(f). In contrast, TGNN often misinterprets sentences and is easily misled by distractors. These qualitative results highlight RefMask3D's superior ability to fuse and align vision-language features, enabling a deeper understanding of complex language descriptions.

## 5 CONCLUSION

In this work, we present an effective single-stage approach **RefMask3D** to address the challenging 3D referring segmentation and grounding. In the proposed RefMask3D, first, a Geometry-Enhanced Group-Word Attention (GEGWA) is introduced for continuous and contextual fusion of language and vision features at the encoding stage. Then, a Linguistic Primitives Construction (LPC) is utilized to learn semantic primitives for representing distinct semantic attributes, enhancing fine-grained vision-language understanding at decoding stage. Moreover, an Object Cluster Module (OCM) is designed to capture holistic information and produce object embedding for generating the final segmentation prediction. The proposed RefMask3D consistently achieves new state-of-the-art performance on 3D referring segmentation, 3D visual grounding, and 2D referring image segmentation. Especially, RefMask3D surpasses the previous 3D referring segmentation methods by a large margin of **5.36% mIoU** on ScanRefer dataset, demonstrating its effectiveness.

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
