# OpenReview forum: "RefMask3D: Language-Guided Transformer for 3D Referring Segmentation"
_acmmm.org/ACMMM/2024/Conference — MM2024 Poster_

### Official Review · Reviewer_LVW9 · 2024-05-27

**Rating:** 6
**Confidence:** 3

**Summary:**

The main content of this paper is the introduction of a novel approach called RefMask3D for3D referring segmentation and grounding. The paper addresses the challenges of fusing and aligning vision-language features, enhancing feature interaction, and understanding complex language descriptions in the context of3D visual scenes.

**Strengths:**

It is very novelty. The paper proposes several novel contributions.  It introduces a Geometry-Enhanced Group-Word Attention mechanism that enhances cross-modal interactions by integrating language with geometrically coherent sub-clouds.  It also presents a Linguistic Primitives Construction strategy to learn primitives for representing distinct semantic attributes.  Additionally, an Object Cluster Module is introduced to analyze the interrelationships among linguistic primitives.  These novel ideas contribute to the improvement of 3D referring segmentation and visual grounding tasks.

**Limitations:**

The paper briefly mentions the performance of the proposed method (RefMask3D) in terms of mIoU (mean Intersection over Union) and Acc@0.5 (accuracy at an IoU threshold of 0.5) on the ScanRefer dataset.  While achieving state-of-the-art performance is mentioned, the paper does not provide a comprehensive evaluation or comparison with other existing methods. For example, 2D-based methods.
Besides, the paper evaluates the proposed method on several benchmark datasets, including ScanRefer.  It compares the performance of RefMask3D with state-of-the-art methods, demonstrating superior results in terms of mIoU and Acc@IoU metrics. The evaluation is comprehensive and provides a fair comparison with existing approaches.

**Suitability:**

3

---

### Official Review · Reviewer_h9vv · 2024-05-29

**Rating:** 5
**Confidence:** 3

**Summary:**

Aiming to handle the 3D referring segmentation task, this paper proposes RefMask3D to explore the comprehensive multi-modal feature interaction and understanding. It contains 3 key points: (1) Geometry-Enhanced Group-Word Attention; (2)  Linguistic Primitives Construction; (3) Object Cluster Module. From the experimental results, it shows that RefMask3D achieves SOTA performance on multiple tasks, including: 3D referring segmentation, 3D visual grounding, and 2D referring image segmentation.

**Strengths:**

- This paper has good organization and clear figures. It is easy for the readers to follow.
 - RefMask3D achieves SOTA performance on multiple tasks: 3D referring segmentation, 3D visual grounding, and 2D referring image segmentation. Furthermore, it outperforms previous SOTA by a large margin of 5.36% mIoU on ScanRefer benchmark.

**Limitations:**

- The primitives contain common attributes such as location, color, and relationship. In the description, some of the primitives may be related to referred objects, rather than the target. Does the primitives of the reference mislead the learning of the target mask?
 - For the one-stage methods, different segmentation backbones may affect the final metrics on ScanRefer. It is suggested that the authors also conduct experiments on segmentation-independent datasets such as ReferIt3D to verify the cross-modal learning ability of the proposed method.

**Suitability:**

3

---

### Official Review · Reviewer_TX1m · 2024-06-03

**Rating:** 4
**Confidence:** 3

**Summary:**

The paper presents RefMask3D, an innovative model for 3D referring segmentation. RefMask3D integrates language with geometrically coherent sub-clouds using a Geometry-Enhanced Group-Word Attention mechanism and introduces a Linguistic Primitives Construction and Object Cluster Module. The paper provides impressive experimental results that fully validate the effectiveness of the proposed method, achieving SOTA performance. However, the paper lacks some experiments that I would have liked to see, and some of the proposed structures do not demonstrate significant novelty, leaning more towards engineering and the integration of existing methods. Nonetheless, I will still give it a borderline accept score.

**Strengths:**

1. The proposed Geometry-Enhanced Group-Word Attention mechanism and Linguistic Primitives Construction module offer effective methods for vision-language feature fusion and alignment.
2. The paper provides detailed experiments and ablation studies that validate the contributions of each component of the model.

**Limitations:**

My first concern is whether the proposed method will significantly increase the computational burden? Beside, for the 3D visual grounding task, the paper lacks comparisons with some advanced methods, such as 3D-VisTA, which shows better performance across multiple metrics compared to the proposed method. For the 2D tasks, the paper does not include comparisons with advanced methods based on SAM. Additionally, I am curious about the potential effectiveness of using advanced 3D visual grounding methods to detect the bounding box of the referring object and then finding the point cloud within that box.

**Suitability:**

3

---

### Meta-Review · Area_Chair_Pv5s · 2024-07-05

**Recommendation:** Accept (Poster)
**Confidence:** 5

**Metareview:**

After considering all reviews, the rebuttal, and the subsequent discussion, the consensus is to accept the paper.